# How to Promote Logistics Enterprises to Participate in Reverse Emergency Logistics: A Tripartite Evolutionary Game Analysis

**Yumei Luo \*, Yuke Zhang and Lei Yang**

School of Business and Tourism Management, Yunnan University, Kunming 650106, China
\* Correspondence: luoyumei@ynu.edu.cn

**Abstract:** Considering the emergency risks and uncertainties of emergency recycling processes, this research builds a tripartite evolutionary game model of government, logistics enterprises, and environmental non-governmental organizations (NGOs) to study the interaction mechanism. Based on the analysis of evolutionary stable strategy (ESS), this research uses MATLAB R2018b to mainly show the strategy choice trends of logistics enterprises in various scenarios including "Government Failure", as well as the mutual impacts of government and environmental NGOs' strategy selection. The research found that (1) the government has an important role in efficiently promoting logistics enterprises' participation; (2) the net benefits of logistics enterprises and environmental NGOs, as key factors that directly affect the game results, are influenced by emergency risks and uncertainty, respectively; (3) environmental NGOs not only play an effective complementary role to government functions, including in the "Government Failure" context, but can also urge the government to perform regulatory functions. This research enriches the study in the field of the combination of evolutionary game theory and reverse emergency logistics as well as providing a reference for the government in developing economic and administrative policies to optimize the recycling and disposal of emergency relief.

**Keywords:** evolutionary game; environmental NGOs; emergency recycling; logistics enterprises; reverse emergency logistics

## 1. Introduction

### 1.1. Background

In recent years, emergencies such as earthquakes, fires, explosions, hurricanes, SARS, avian flu, and COVID-19 have occurred one after another, posing great threats to people's health and lives, stable social and economic development, the implementation of national strategic planning, and other aspects [1]. The timely and fast arrival of emergency relief to disaster areas is crucial to reduce the losses caused by disasters, especially the secondary injuries after disasters, which has been the focus of existing research [2]. However, emergency events have caused the need for a large number of emergency supplies in a short time, which seriously impacts the balance of supply and demand [2]. Large amounts of excess and expired emergency supplies are generated after emergency events. For example, in 2003, the Xiaotangshan Hospital in Beijing, China, urgently procured about RMB 90 million of emergency relief in aid to fight SARS [3]. However, several years later, great parts of that, including drugs, food, and medical equipment, were still stored in temporary warehouses [3]. In 2014, Australia and New Zealand dumped AUD 200 million and NZD 110 million worth of expired emergency relief, respectively [4]. In May 2008, Wenchuan (a county of Sichuan province in China) received a large number of emergency relief items through different means after an earthquake, some of them even exceeding the needs of the affected areas. Due to a lack of recycling, plenty of food supplies rotted in warehouses, and other damaged emergency relief items were not disposed of until four years after the earthquake [5], which leads to social and economic problems as well as environmental pollution.

Therefore, to reduce waste, protect the environment, and further improve the utilization rate of emergency relief, it is necessary to carry out effective reverse emergency logistics.

## 1.2. Involving Logistics Enterprises and Environmental NGOs

In most emergencies, the government is in charge of emergency logistics in many countries. For example, the Chinese emergency management system adopts a government-led governance model [6]. However, for emergency subjects other than the government, including enterprises and non-governmental organizations (NGOs), there is currently no provision on the method and responsibility of participating in emergency events. The single governance model of government has many disadvantages, such as the fact that enterprises and non-governmental organizations (NGOs) generally lack the awareness of public crisis, insignificant role and efficiency, insufficient ability to deal with emergencies, a lack of multi-subject coordination and cooperation mechanism, and imperfect information sharing mechanisms [6]. Among the multiple subjects of emergencies, the government is the core subject, while enterprises, NGOs, the public and the media are the important forces of the multiple subjects. Some studies found that the model of government-led and multi-participation by enterprises, social organizations, etc., can maximize the combined governance advantages of multi-subject participation and multi-resource joint play [7]. Thus, it is urgent to improve the government-led and multi-subject governance model of reverse emergency logistics.

Some researchers have proposed the benefits of having third-party logistics enterprises assume the responsibility for reverse logistics [8,9]. Spicer and Johnson [8] compared three models of reverse logistics (e.g., OEM Takeback, Pooled Takeback, and Third-Party Takeback) and found that recycling conducted by a third party had the highest efficiency. Chen et al. [9] pointed out that outsourcing reverse logistics to a third party has higher flexibility and practicability. Some studies have found that compared with the producer recycling model, third-party recycling can make the recycling process professional and standardized to improve recycling efficiency [10]. Therefore, it is inevitable that the government to introduce logistics enterprises into its emergency recycling processes.

Nevertheless, no other entities involved, except the government and logistics enterprises, are able to induce "Government Failure", which entails a lack of professional technical knowledge [11], buck-passing among officials, and enterprises bribing relevant government departments. If "Government Failure" cannot be alleviated in time, a series of problems caused by it will bring about various counterproductive results in the emergency recycling processes, which will directly lead to the low efficiency of emergency recycling and damage to social welfare [12]. NGOs are the third sector of emergency events between enterprises and the government. They are independent, non-profit and voluntary public welfare social organizations [13] and have become a social supervision force that cannot be ignored outside the government [14]. Gourevitch et al. [15] proposed that the intervention of professional NGOs in relevant fields could effectively alleviate "Government Failure". Tan et al. [16] argue that NGOs' intervention can not only improve the effect of the government's regulation, but also supervise the government's moral behavior; it has become an important remedy for "Government Failure". As a result, in terms of emergency recycling, environmental NGOs should also be involved.

Until now, Chinese reverse emergency logistics development has been in its infancy, and relevant institutions and regulations are incomplete. As mentioned above, logistics enterprises and environmental NGOs must be involved in emergency recycling processes, which depend on the strategic interaction between the supervision of government regulators and the operating behaviors of logistics enterprises and environmental NGOs. As a result, the government, logistic enterprises, and environmental NGOs naturally form a competitive and cooperative relationship when it comes to emergency recycling processes [17]. Obviously, it is an aspect of multi-party game problems, while evolutionary game theory is a widely useful tool to analyze those problems [18,19]. Considering the decision-making processes of the government, logistics enterprises, and environmental

NGOs, this research build a tripartite evolutionary game model under the assumption of bounded rationality. Additionally, we also explore and analyze the dynamic evolutionary mechanism between those three game players, thereby improving the emergency recycling processes and making the game system evolve in a direction favorable to all participants. Based on that, this paper attempts to propose a theoretical basis as well as a practical reference for the government in developing economic and administrative policies to optimize the recycling and disposal of emergency relief.

## 2. Literature Review

### 2.1. Emergency Logistics and Reverse Emergency Logistics

At present, the research of emergency logistics focuses on distribution, evaluation and optimization, and organization management. Reverse emergency logistics is an important part of emergency logistics but now it obtains less attention.

(1) The distribution of emergency relief

To find the best assignment of available resources to operational area, Fiedrich et al. [20] constructed a dynamic optimization model from the perspective of reducing logistics investment. Chen et al. [21] built a multi-commodity, multi-modal, multi-period, and multi-objective model to deal with the problems of facility location and emergency materials allocation. Taking the safety factor, time factor, and economic factor, Guo and Wang [22] introduced an improved Dijkstra algorithm and built a model of emergency logistics to analyze the path of complex water area. Zhang [23] analyzed synergistic effect among the emergency rescue guarantee departments through referring to the dynamic allocation Multi-agent theory. In facilitating and improving the scheduling of emergency resources, Bodaghi et al. [24] generated a model integrates GIS and Mixed Integer Programming (MIP) approaches.

(2) The evaluation and optimization of emergency logistics systems

Focusing on optimizing the selection mechanism of facility location decisions, as for an emergency relief chain, Balcik et al. [25] developed a new model on the basis of maximal covering location model. Jeong et al. [26] presented an integrated framework for designing emergency logistics networks (ELNs) that considers efficiency, risk, and robustness. Lu et al. [27] established an information integration model based on metadata to figure out the problem that distributed, heterogeneous emergency information collaboration and share difficultly. Yu et al. [28] considered the impact of time on the location of emergency facilities and established a model aimed at site configuration optimization with the goal of time satisfaction. Concentrated on COVID-19, Zhang et al. [29] pointed out the shortcomings of China's emergency logistics system and formed a targeted emerging emergency logistics system on this basis. Combining this with the theory of system dynamics, Liu et al. [30] also aimed at COVID-19 to construct mathematical models for four dynamic links, thereby dynamically optimizing the decision-making framework model of emergency logistics networks.

(3) The organization and management of emergency logistics

Considering the relationship between the different departments in the rescue process, Pettit et al. [31] built the emergency logistics demand model. Hu et al. [32] in view of the coordination of information, resources and organization, generated the resilience model of emergency logistics network by using the system dynamics. To improve the participation of commercial organizations, L'Hermitte et al. [33] developed a theoretical framework to analyzing the structure, benefits, and prerequisites of a logistics-sharing system. Peterson et al. [34] proved that key elements of supply chain theory could optimized the management of emergency logistics, through a series of case studies.

(4) Reverse emergency logistics

In order to improve the decision-making of reverse emergency logistics, Guo et al. [35] built a system dynamic model to simulate the recycle process. Hu et al. [36] proposed a reverse logistics system of post-disaster debris. Aiming at the medical waste conducted by COVID-19, Mei et al. [37] established a multi-period medical waste emergency reverse

logistics network. Wang et al. [38] considered the demand of people in disaster area and introduced forward and reverse logistics to construct a Location-Routing Problem model of emergency facilities. According to Stochastic Petri Net (SPN) and involving reverse logistics, Yu et al. [39] generated an emergency resource allocation model.

Actually, the above research utilized various methods, but most of them conducted analyses from a static perspective. This research lens is unable to fully capture the dynamic aspects of the linkage development process or to reveal the evolutionary law of the game relationship between the participants as they strive to maximize their individual interests. Furthermore, those literature mainly focus on forward emergency logistics, while ignoring the problems of reverse emergency logistics to some extent. Tomasini et al. [40] pointed out that in the context of emergency management, more attention is paid to how to deliver critical supplies to the relevant areas efficiently, so that to meet the needs of those affected by disasters or other emergency events. Peretti et al. [41] had also argued that the lack of "profit attractiveness" has become the main barrier of reverse emergency logistics in the processes of development. Hence, it is necessary to analyze the multi-participant emergency recycling mechanism from a dynamic perspective.

### 2.2. Evolutionary Game Theory

In the 1950s, Herbert A. Simon, the pioneer of behavioral decision theory, published a book entitled "Administrative Behavior" [42]. In it, he argued that, impacted by perceptual bias (relying on intuition more than logic), the limitations of objective resources, personal capability, and attitudes to risk, decision-makers cannot make the most optimal decisions in extreme, uncertain, and complicated situations. Evolutionary game theory is generated with the combination of game theory and evolutionary concepts. It considers the bounded rationality and incomplete information of all parties in the game and could describe the dynamic nature of the participants' strategy evolution process comprehensively [43]. In recent years, evolutionary game theory has been widely used in the research of reverse logistics.

Gu et al. [44] built a two-party evolutionary game model to investigate the cooperation relationship of NREI (natural resource- and energy-intensive industries) companies which use self-operation or joint venture reverse logistics operating strategy. Focusing on the WEEE (waste electrical and electronic equipment) recycling, Zhao et al. [45] established evolutionary game models and analyzed strategies evolution process of producers and recyclers under two different fund policies. To explore the major influencing factors of reverse logistics in e-commerce, Gong et al. [46] established an evolutionary game model of local country government and third-party logistics enterprises to discuss the major influencing factors of government–enterprise collaboration under COVID-19. He et al. [47] built an evolutionary game model of e-commerce companies and consumers on the reverse logistics of express packing under government regulations. Li et al. [48] generated a tripartite evolutionary game model of government, pharmaceutical companies, and consumers for the reverse logistics of drugs. These studies do contribute to the advancement of research on the integration of evolutionary game theory and reverse logistics. However, most of them merely construct evolutionary game models to analyze the interaction of enterprises, the government, and customers, while ignoring the involvement of environmental NGOs and issues involved in reverse emergency logistics.

As previously mentioned, environmental NGOs could take part in the government and logistics enterprises' binary emergency recycling mechanism as the impartial third-party force. In this way, environmental NGOs can further exert pressure on logistics enterprises and increase their participation while making up for the government's shortcomings. Obviously, there must be complexity and uncertainty in the emergency recycling mechanism of 'Government–Logistics Enterprises–Environmental NGOs'. First, logistics enterprises will face the risks of emergency events, such as the risk of secondary infection in large-scale epidemics, aftershocks, and other similar risks, and the occurrence of these risks is uncertain. Second, the NGOs must also deal with the uncertainties brought by the emergency situation and other emergency recycling mechanism participants' unpredictable behaviors [49,50].

As a result, using evolutionary game theory to study their interaction under the premise of bounded rationality will be beneficial in solving the problem of reverse emergency logistics and filling the gap in the field of the combination of evolutionary game theory and reverse emergency logistics.

## 3. The Model

### 3.1. Problem Formulations

According to the emergency recycling mechanism of this study, logistics enterprises undertake the main recycling work, while the government and environmental NGOs play the roles of guidance, supervision, and financial support. Therefore, this paper focuses more on how to motivate logistics enterprises to actively participate in the recycling process.

Considering that the costs of logistics enterprises and NGOs will be influenced by emergency risks and uncertainties, respectively, in the process of participating in emergency recycling, this study further quantifies the degree of emergency risks and uncertainties to propose a tripartite evolutionary game model to analyze the participants' strategy evolution. Data stimulation via MATLAB R2018b was applied to further discuss the variables influencing logistics enterprises' strategy choice and to propose certain recommendations.

### 3.2. Basic Assumptions of the Model

#### 3.2.1. Strategy Selection of Players

As mentioned above, there are three players in the evolutionary game: the government, logistics enterprises, and environmental NGOs. In order to analyze the problem, this paper summarizes the basic settings of strategy selection for those players:

(1) This study proposes the government can choose two strategies: "reward-punishment" and "no reward-punishment". "Reward-punishment" refers to the government actively providing awards to logistics enterprises and environmental NGOs that actively participated in the takeback and recycling processes, and leveraging taxes to punish the logistics enterprises that do not participate in recycling. "No reward-punishment" refers to the government taking no action in recycling processes. As a result, it will pay certain costs.

(2) The strategies of logistics enterprises are divided into "participation" and "non-participation". "Participation" indicates enterprises which actively participate in the reverse logistics of emergency relief. "Non-participation" refers to logistics enterprises which only conduct regular takeback and recycling instead of emergency recycling, and they need to pay the taxes levied by the "reward-punishment" government.

(3) Environmental NGOs' strategies include "participation" and "non-participation". "Participation" refers to fully independent third-party environmental NGOs which actively supervise the engagement of the government and logistics enterprises in the recycling processes and query the "no reward-punishment" government and "non-participation" logistics enterprises through public opinion and public interest litigation. "Non-participation" refers to the organizations which take no action in emergency recycling.

#### 3.2.2. Assumptions of Players

Elements of the evolutionary game are set as follows:

**Assumption 1.** *At the beginning of the game, the probability that the government chooses "reward-punishment" is x (hence the probability of the no reward-punishment is $(1 - x)$); the probability that logistics enterprises select the "participation" is y (hence the probability of non-participation is $(1 - y)$); and the probability that environmental NGOs choose "participation" is z (hence the probability of no participation is $(1 - z)$), where $0 \leq x$, $y$, $z \leq 1$*

**Assumption 2.** *When the "reward-punishment" strategy is chosen, the government will bear certain administrative costs $C_{G1}$, provide logistics enterprises with subsidies $C_{G2}$ and reward NGOs with $C_{G3}$. When logistics enterprises adopt the "participation" strategy, the government will*

*enjoy burden relief and then gain indirect benefits $R_{G1}$. In the same way, when the "participation" strategy is adopted by environmental NGOs, the government will save costs and then gain indirect benefits $R_{G2}$. Whereas, when the "non-participation" strategy is selected by logistics enterprises, the government will encounter heavier burdens and then pay indirect costs $C_{G4}$, and the logistics enterprises need to pay taxes $R_{G3}$. When the government adopts the "no reward-punishment" strategy, the supervision conducted by the environmental NGOs choosing the "participation" strategy will cause potential losses $\Delta C$ to the government.*

**Assumption 3.** *When the "non-participation" strategy is selected, logistics enterprises will gain revenue $R_{E1}$ with regular cost $C_{E1}$. In this situation, with the public credibility decreasing and social image damaged, logistics enterprises will pay indirect costs $C_{E2}$, and taxes $R_{G3}$ levied by the "reward-punishment" government. Furthermore, the environmental NGOs choosing the "participation" strategy will also inquire the 'non- participation' logistics enterprises via its supervision function and cause extra losses $\Delta C_{E2}$ to them. Logistics enterprises selecting the "participation" strategy will gain revenue $R_{E2}$ through re-sale and re-use of emergency relief and receive government subsidies $C_{G2}$. Moreover, besides regular cost $C_{E1}$, logistics enterprises selecting the "participation" strategy will face emergency risks. Hence the impact of emergency risks on logistics enterprises' strategy selection is quantified as a risk cost coefficient $\beta$ ($0 < \beta < 1$).*

**Assumption 4.** *When the "non-participation" strategy is selected, environment NGOs will gain regular financial support $R_{S1}$ with paying regular supervision cost $C_{S1}$. When the "participation" strategy is adopted, environmental NGOs can gain financial support $R_{S2}$ under emergency and government rewards $C_{G3}$. Moreover, except for regular operation costs $C_{S1}$, environmental NGOs selecting the "participation" strategy will also face uncertainties in emergency context. Therefore, the impact of emergency uncertainties on environmental NGOs' strategy selection is quantified as an uncertainty cost coefficient $\gamma$ ($0 < \gamma < 1$).*

### 3.2.3. Income Matrix

Based on the above assumptions, the three-party evolutionary game is described by the matrix in Table 1.

**Table 1.** Evolutionary game income matrix.

| Government and Logistics Enterprises | | Environmental NGOs | |
|---|---|---|---|
| | | **"Participation" ($z$)** | **"Non-Participation" ($1-z$)** |
| **Government "reward-punishment" ($x$)** | **Logistics enterprises "participation" ($y$)** | $R_{G1} + R_{G2} - C_{G1} - C_{G2} - C_{G3}$ <br> $R_{E2} + C_{G2} - (1+\beta)C_{E1}$ <br> $R_{S2} + C_{G3} - (1+\gamma)C_{S1}$ | $R_{G1} - C_{G1} - C_{G2}$ <br> $R_{E2} + C_{G2} - (1+\beta)C_{E1}$ <br> $R_{S1} - C_{S1}$ |
| | **Logistics enterprises "non-participation" ($1-y$)** | $R_{G2} + R_{G3} - C_{G1} - C_{G3} - C_{G4}$ <br> $R_{E1} - C_{E1} - C_{E2} - \Delta C_{E2} - R_{G3}$ <br> $R_{S2} + C_{G3} - (1+\gamma)C_{S1}$ | $R_{G3} - C_{G1} - C_{G4}$ <br> $R_{E1} - C_{E1} - C_{E2} - R_{G3}$ <br> $R_{S1} - C_{S1}$ |
| **Government "no reward-punishment" ($1-x$)** | **Logistics enterprises "participation" ($y$)** | $R_{G1} - C$ <br> $R_{E2} - (1+\beta)C_{E1}$ <br> $R_{S2} - (1+\gamma)C_{S1}$ | $R_{G1}$ <br> $R_{E2} - (1+\beta)C_{E1}$ <br> $R_{S1} - C_{S1}$ |
| | **Logistics enterprises "non-participation" ($1-y$)** | $-C_{G4} - \Delta C$ <br> $R_{E1} - C_{E1} - C_{E2} - \Delta C_{E2}$ <br> $R_{S2} - (1+\gamma)C_{S1}$ | $-C_{G4}$ <br> $R_{E1} - C_{E1} - C_{E2}$ <br> $R_{S1} - C_{S1}$ |

Note: the benefit function of each column in the table is government income, logistics enterprises income, environmental NGOs income.

### 3.3. Establishment of the Tripartite Evolutionary Game Model

When a participant makes decisions, evolutionary game theory holds that the party with a low-benefit strategy will evolve to a high-benefit strategy with time. Thus, the number of people making corresponding decisions in the decision-making group will

continue to change. To reveal and describe the evolution process of participant's strategic choice, the Replicator Dynamic Equation (RDE) is proposed. When the RDEs of each party are simultaneous and the solution is set as '0', an equilibrium can be obtained, namely the strategy combinations. Further analysis is needed to discuss whether these equilibriums are the Evolutionary Stable Strategy (ESS) of this game system.

According to the income matrix, when the government chooses "reward-punishment", its expected income is $G_{R1}$; when the government chooses "reward-punishment", its expected income is $G_{R2}$. The average income of the government is $G_R$.

$$G_{R1} = -C_{G1} - yC_{G2} - zC_{G3} - C_{G4} + yC_{G4} + yR_{G1} + zR_{G2} + R_{G3} - yR_{G3}, \tag{1}$$

$$G_{R2} = -z\Delta C + (-1+y)C_{G4} + yR_{G1}, \tag{2}$$

$$G_R = x \times G_{R1} + (1-x)G_{R2}. \tag{3}$$

When the logistics enterprises choose "participation", their expected income is $E_{R1}$; when the logistics enterprises choose "non-participation", their expected income is $E_{R2}$. The average income of the logistics enterprises is $E_R$.

$$E_{R1} = -(1+\beta)C_{E1} + xC_{G2} + R_{E2}, \tag{4}$$

$$E_{R2} = -C_{E1} - C_{E2} + R_{E1} - xR_{G3} - z\Delta C_{E2}, \tag{5}$$

$$E_R = y \times E_{R1} + (1-y)E_{R2}. \tag{6}$$

When the environmental NGOs choose "participation", their expected income is $S_{R1}$; when the environmental NGOs choose "non-participation", their expected income is $S_{R2}$ The average income of the environmental NGOs is $S_R$.

$$S_{R1} = xC_{G3} - (1+\gamma)C_{S1} + R_{S2}, \tag{7}$$

$$S_{R2} = -C_{S1} + R_{S1}, \tag{8}$$

$$S_R = z \times S_{R1} + (1-z)S_{R2}. \tag{9}$$

The replication dynamic equation is a dynamic differential equation that describes the frequency at which a particular strategy is employed in a population. The equation for the government selecting "reward-punishment" is:

$$F(x) = \frac{dx}{dt} = x(G_{R1} - G_R) = x(1-x)[-C_{G1} + z(\Delta C - C_{G3} + R_{G2}) + R_{G3} - y(C_{G2} + R_{G3})]. \tag{10}$$

Similarly, the replication dynamic equations of the logistics enterprise and environmental NGOs that choose "participation" are, respectively,

$$F(y) = \frac{dy}{dt} = y(E_{R1} - E_R) = y(1-y)[-\beta C_{E1} + C_{E2} - R_{E1} + R_{E2} + x(C_{G2} + R_{G3}) + z\Delta C_{E2}]. \tag{11}$$

$$F(z) = \frac{dz}{dt} = z(S_{R1} - S_R) = z(1-z)(xC_{G3} - \gamma C_{S1} - R_{S1} + R_{S2}). \tag{12}$$

When $F(x) = 0$, $F(y) = 0$ and $F(z) = 0$, there are nine equilibrium solutions in the tripartite evolutionary game, which are, respectively, (0,0,0), (0,0,1), (0,1,0), (0,1,1), (1,0,0), (1,0,1), (1,1,0), (1,1,1), ($x^*, y^*, z^*$), where

$$x^* = \frac{\gamma C_{S1} + R_{S1} - R_{S2}}{C_{G3}},$$

$$y^* = \frac{z\Delta C - C_{G1} - zC_{G3} + zR_{G2} + R_{G3}}{C_{G2} + R_{G3}},$$

$$z^* = \frac{\beta C_{E1} - C_{E2} - xC_{G2} + R_{E1} - R_{E2} - xR_{G3}}{\Delta C_{E2}}.$$

## 4. Analysis of Evolutionary Stable Strategy (ESS)

*4.1. ESS Analysis of Game Players*

According to the evolutionary game theory, when $F'(x) < 0$, $F'(y) < 0$ and $F'(z) < 0$, a stable equilibrium is reached, where $x^*$, $y^*$, $z^*$ are the evolutionary stability points of the government, logistics enterprises and environmental NGOs.

(1) Evolutionary stability analysis of the government

$$F'(x) = (1 - 2x)[-C_{G1} + z(\Delta C - C_{G3} + R_{G2}) + R_{G3} - y(C_{G2} + R_{G3})]. \qquad (13)$$

When $y = y^*$, $\forall x$ makes $F(x) = 0$, $x \in (0, 1)$.

When $y \neq y^*$, we can obtain two possible evolutionary stable points with $F(x) = 0$ at $x = 1$ and $x = 0$;

Let $F'(x) < 0$, then

(1) If $y < y^*$, then $F'(0) > 0$ and $F'(1) < 0$, which means the evolutionary stable point is $x = 1$, which indicates that when the probability of logistics enterprises implementing reverse logistics is low, the government tends to choose the "reward-punishment" strategy considering the increased burden of emergency follow-up work.

(2) If $y > y^*$, then $F'(0) < 0$ and $F'(1) > 0$, which means the evolutionary stable point is $x = 0$, which indicates that when the probability of logistics enterprises choosing the "participation" strategy is high, then even without government subsidies, logistics enterprises will actively take the takeback and recycling affairs; then, the government tends to choose the "no reward-punishment" strategy.

(2) Evolutionary stability analysis of logistics enterprises

$$F'(y) = (1 - 2y)[-\beta C_{E1} + C_{E2} - R_{E1} + R_{E2} + x(C_{G2} + R_{G3}) + z\Delta C_{E2}]. \qquad (14)$$

When $z = z^*$, $\forall y$ makes $F(y) = 0$, $y \in (0, 1)$.

When $z \neq z^*$, we can obtain two possible evolutionary stable points with $F(x) = 0$ at $y = 1$ and $y = 0$:

Let $F'(y) < 0$, then

(1) If $z < z^*$, then $F'(0) < 0$ and $F'(1) > 0$, which means the evolutionary stable point is $y = 0$, which indicates that when the probability of environmental NGOs to choose the "participation" strategy is low, logistics enterprises will not bear the indirect costs caused by NGOs' supervision and will gain benefits through regular recycling affairs. Considering emergency risks and uncertainties, the cost of emergency recycling processes is uncertain, and enterprises are unable to confirm whether the cost will be covered by benefits or not. Therefore, logistics enterprises tend to choose the "non-participation" strategy.

(2) If $z > z^*$, then $F'(0) > 0$ and $F'(1) < 0$, which means the evolutionary stable point is $y = 1$, which indicates that when the probability of environmental NGOs to choose the "participation" strategy is high, logistics enterprises will bear the indirect cost caused by NGOs' supervision, if they choose the 'non- participation' strategy. Then, the cost is expected to be larger than its benefits. After that, logistics enterprises tend to choose the 'implement' strategy.

(3) Evolutionary stability analysis of environmental NGOs

$$F'(z) = (1 - 2z)(xC_{G3} - \gamma C_{S1} - R_{S1} + R_{S2}). \qquad (15)$$

When $x = x^*$, $\forall z$ makes $F(z) = 0$, $z \in (0, 1)$.

When $x \neq x^*$, we can obtain two possible evolutionary stable points with $F(z) = 0$ at $z = 1$ and $z = 0$:

Let $F'(z) < 0$, then:

(1)  If $x < x^*$, then $F'(0) < 0$ and $F'(1) > 0$, which means the evolutionary stable point is $z = 0$, which indicates that when the probability of government choosing the "reward-punishment" strategy is low, environmental NGOs cannot receive government subsidies and supervision cost cannot be covered by the benefits considering emergency risks. Therefore, environment NGOs will choose the 'non-participation' strategy.

(2)  If $x > x^*$, then $F'(0) > 0$ and $F'(1) < 0$, which means the evolutionary stable point is $z = 1$, which indicates that when the probability of government choosing the "reward-punishment" strategy is high, environmental NGOs can receive government subsidies and its supervision costs can be covered by its benefits considering emergency risks. Therefore, environmental NGOs will choose the "participation" strategy.

*4.2. ESS Analysis of Three-Party Evolutionary Game*

4.2.1. ESS Analysis of the Government and Logistics Enterprises

The variables related to environmental NGOs are taken as constants to analyze the ESS of the other two game players.

According to the RDE of the government and logistics enterprises, when and only when $0 \le \frac{\beta C_{E1} - C_{E2} + R_{E1} - R_{E2} - z\Delta C_{E2}}{C_{G2} + R_{G3}} \le 1, 0 \le \frac{z\Delta C - C_{G1} - zC_{G3} + zR_{G2} + R_{G3}}{C_{G2} + R_{G3}} \le 1$, the Jacobian matrix of this system is yielded as:

$$J_1 = \begin{pmatrix} \frac{\sigma F(x)}{\sigma x} & \frac{\sigma F(x)}{\sigma y} \\ \frac{\sigma F(y)}{\sigma x} & \frac{\sigma F(y)}{\sigma y} \end{pmatrix}$$

$$= \begin{pmatrix} (1-2x)(-C_{G1} + z(\Delta C - C_{G3} + R_{G2}) + R_{G3} - y(C_{G2} + R_{G3})) & -(1-x)x(C_{G2} + R_{G3}) \\ (1-y)y(C_{G2} + R_{G3}) & (1-2y)(-\beta C_{E1} + C_{E2} - R_{E1} + R_{E2} + x(C_{G2} + R_{G3}) + z\Delta C_{E2}) \end{pmatrix}. \quad (16)$$

According to evolutionary game theory, for matrixes with different equilibrium points, if $DET(J_i) > 0$ and $TR(J_i) < 0$ $(i = 1,2)$, then the stable equilibrium points of the evolutionary system can be obtained. Moreover, according to the linear algebra theory, the specific determinants and trace can be determined with the eigenvalues corresponding to these equilibrium points. Based on that, we find that the evolutionary game of logistics enterprises and the government finally has two evolutionary game stability points, which are (1,0) and (1,1), respectively (see Table 2). Since $(x^*, y^*)$ is not the stable point of evolutionary game of enterprises and governments in any case, a detailed discussion will not be made.

**Table 2.** Matrix eigenvalues ($\mu_1$, $\mu_2$) corresponding to the equilibrium points.

| | $\mu_1$ | $\mu_2$ |
|---|---|---|
| $x = 0, y = 0$ | $z\Delta C - C_{G1} - zC_{G3} + zR_{G2} + R_{G3}$ | $-\beta C_{E1} + C_{E2} - R_{E1} + R_{E2} + z\Delta C_{E2}$ |
| $x = 0, y = 1$ | $z\Delta C - C_{G1} - C_{G2} - zC_{G3} + zR_{G2}$ | $\beta C_{E1} - C_{E2} + R_{E1} - R_{E2} - z\Delta C_{E2}$ |
| $x = 1, y = 0$ | $-z\Delta C + C_{G1} + zC_{G3} - zR_{G2} - R_{G3}$ | $-\beta C_{E1} + C_{E2} + C_{G2} - R_{E1} + R_{E2} + R_{G3} + z\Delta C_{E2}$ |
| $x = 1, y = 1$ | $-z\Delta C + C_{G1} + C_{G2} + zC_{G3} - zR_{G2}$ | $\beta C_{E1} - C_{E2} - C_{G2} + R_{E1} - R_{E2} - R_{G3} - z\Delta C_{E2}$ |

Note: The product of $\mu_1$ and $\mu_2$ is the determinant (DET) of the Jacobian matrix $J_1$; the sum of $\mu_1$ and $\mu_2$ is the trace (TR) of the Jacobian matrix $J_1$.

When $(C_{G1} + zC_{G3}) - (zR_{G2} + R_{G3}) < z\Delta C$, namely the net loss of government selecting the "reward-punishment" strategy is less than that caused by NGOs when the government selects the "no reward-punishment" strategy, and $R_{E2} + xC_{G2} - (1 + \beta)C_{E1} < R_{E1} - C_{E1} - C_{E2} - z\Delta C_{E2} - xR_{G3}$, namely the logistics enterprises' benefits of reverse emergency logistics are lower than regular reverse logistics, the equilibrium point of both parties is (1,0). In other words, in this situation, the government chooses reward-punishment, while logistics enterprises do not implement reverse logistics.

When $C_{G1} + C_{G2} + zC_{G3} - zR_{G2} < z\Delta C$, namely the net loss of government selecting the "reward-punishment" strategy is less than that caused by NGOs when the government selects the "no reward-punishment" strategy, and $R_{E2} + xC_{G2} - (1 + \beta)C_{E1} > R_{E1} - C_{E1} - C_{E2} - z\Delta C_{E2} - xR_{G3}$, namely the logistics enterprises' benefits of emergency reverse logistics is higher than regular reverse logistics, the equilibrium point of both

parties is (1,1). In other words, the government chooses the "reward-punishment" strategy and logistics enterprises also choose to implement reverse logistics.

### 4.2.2. ESS Analysis of the Government and Environmental NGOs

The variables related to logistics enterprises are taken as constants to analyze the ESS of the other two game players.

According to the RDE of government and logistics enterprises, when and only when $0 \leq \frac{\gamma C_{S1} + R_{S1} - R_{S2}}{C_{G3}} \leq 1, 0 \leq \frac{C_{G1} + yC_{G2} - R_{G3} + yR_{G3}}{\Delta C - C_{G3} + R_{G2}} \leq 1$, the Jacobian matrix of the system $J_2$ is yielded as:

$$
J_2 = \begin{pmatrix} \frac{\sigma F(x)}{\sigma x} & \frac{\sigma F(x)}{\sigma z} \\ \frac{\sigma F(z)}{\sigma x} & \frac{\sigma F(z)}{\sigma z} \end{pmatrix}
$$
$$
= \begin{pmatrix} (1 - 2x)(-C_{G1} + z(\Delta C - C_{G3} + R_{G2}) + R_{G3} - y(C_{G2} + R_{G3})) & (1 - x)x(\Delta C - C_{G3} + R_{G2}) \\ (1 - z)zC_{G3} & (1 - 2z)(xC_{G3} - \gamma C_{S1} - R_{S1} + R_{S2}) \end{pmatrix}. \tag{17}
$$

According to evolutionary game theory, for matrixes with different equilibrium points, if $DET(J_i) > 0$ and $TR(J_i) < 0$ ($i = 1,2$), then the stable equilibrium points of the evolutionary system can be obtained. Moreover, according to the linear algebra theory, the specific determinants and trace can be determined with the eigenvalues corresponding to these equilibrium points. Based on that, we find that the evolutionary game of the government and environmental NGOs finally has two evolutionary game stability points, which are (0,0) and (1,1), respectively (see Table 3). Since $(x^*, z^*)$ is not the stable point of evolutionary game of enterprises and governments in any case, a detailed discussion will not be made.

**Table 3.** Matrix eigenvalues ($h_1$, $h_2$) corresponding to the equilibrium points.

| | $h_1$ | $h_2$ |
|---|---|---|
| $x = 0, z = 0$ | $-C_{G1} - yC_{G2} + R_{G3} - yR_{G3}$ | $-\gamma C_{S1} - R_{S1} + R_{S2}$ |
| $x = 0, z = 1$ | $\Delta C - C_{G1} - yC_{G2} - C_{G3} + R_{G2} + R_{G3} - yR_{G3}$ | $\gamma C_{S1} + R_{S1} - R_{S2}$ |
| $x = 1, z = 0$ | $C_{G1} + yC_{G2} - R_{G3} + yR_{G3}$ | $C_{G3} - \gamma C_{S1} - R_{S1} + R_{S2}$ |
| $x = 1, z = 1$ | $-\Delta C + C_{G1} + yC_{G2} + C_{G3} - R_{G2} - R_{G3} + yR_{G3}$ | $-C_{G3} + \gamma C_{S1} + R_{S1} - R_{S2}$ |

Note: The product of $h_1$ and $h_2$ is the determinant (DET) of the Jacobian matrix $J_2$; the sum of $h_1$ and $h_2$ is the trace (TR) of the Jacobian matrix $J_2$.

When $R_{S2} - (1 + \gamma)C_{S1} < R_{S1} - C_{S1} - C_{S2}$, namely for environmental NGOs, the benefits of performing supervision is less than that of not performing in emergency recycling processes, and for the government, when $(1 - y)R_{G3} < C_{G1} + yC_{G2}$ and $C_{G3} - R_{G2} < \Delta C$, namely the benefits of general reward-punishment strategy, are less than its costs, the government will choose the 'no reward-punishment' strategy even if the net loss of motivating NGOs is less than the indirect loss caused by NGO's supervision. The equilibrium point of both parties is (0,0). In other words, in this situation, the government is passive, and environmental NGOs do not perform supervisory functions.

When $R_{S2} + xC_{G3} - (1 + \gamma)C_{S1} > R_{S1} - C_{S1}$, namely for environmental NGOs, the general benefits of implementing supervision are larger than that of not implementing in an emergency; and for the government, when $(C_{G1} + zC_{G3} + yC_{G2}) - [zR_{G2} + (1 - y)R_{G3}] < z\Delta C$, namely the net loss of taking reward-punishment strategy is less than the indirect loss caused by NGO's supervision, then the equilibrium point of both parties is (1,1). In other words, in this situation, the government will choose the "reward-punishment" strategy, and the environmental NGOs will choose to perform supervision functions, that is, the "participation" strategy.

## 5. Data Simulation

In the analysis of the tripartite evolutionary game model, this paper more focuses on how to promote logistics enterprises to participate in the emergency recycling processes. Therefore, based on the above analysis, using MATLAB R2018b, the evolution results of

logistics enterprises which are simulated under the situation of the probabilities of logistics enterprises and environmental NGOs choosing "participation" are low, and their income from participating in emergency recycling is less than the cost ($R_{E2} - (1 + \beta)C_{E1} < 0$ and $R_{S2} - (1 + \gamma)C_{S1} < 0$).

First, considering different degrees of emergency risks ($\beta$), the impact of the participation of government and environmental NGOs on the strategic choice of logistics enterprises was examined, which helped us to verify the role of government and environmental NGOs in promoting logistics enterprises to assume the responsibility of reverse logistics.

Second, considering different degrees of "Government Failure", the influence of environmental NGOs' intervention on the strategic choice of logistics enterprises was analyzed, so as to test whether environmental NGOs could significantly make up for the defects of government's supervision and alleviate "Government Failure" in emergency recycling processes.

Finally, the mutual impacts between government and environmental NGOs are further simulated and analyzed, so as to find out the indirect influences of government and environmental NGOs on the decision-making of logistics enterprises.

As mentioned above, when $R_{E2} - (1 + \beta)C_{E1} < 0$ and $R_{S2} - (1 + \gamma)C_{S1} < 0$, the initial parameter values are as follows:

$R_{E1} = 10$, $R_{E2} = 9$, $C_{E1} = 8$, $C_{E2} = 3$, $\beta = 0.6$, $\Delta C_{E2} = 2$, $C_{G2} = 4$, $R_{G3} = 2$, $C_{G1} = 7$, $\Delta C = 12$, $C_{G3} = 8$, $R_{S1} = 9$, $R_{S2} = 10$, $C_{S1} = 8$, $\gamma = 0.3$ and $y = 0.2$, $x = 0.5$, $z = 0.2$.

*5.1. Impact of Government and Environmental NGOs' Participation on the Strategic Selection of Logistics Enterprises*

As for the tripartite evolutionary game model of government, logistics enterprises, and environmental NGOs, the emergency risks faced by logistics enterprises was quantified as a risk cost coefficient $\beta$ ($0 < \beta < 1$). In this section, with initial values setting, this paper makes $\beta = 0.3, 0.6, 0.9$ to indicate the low, medium, and high degree risks faced by logistics enterprises when they undertake reverse emergency logistics.

**Scenario 1.** *The government and environmental NGOs take a neutral attitude to participating in emergency recycling processes, and logistics enterprises face different degrees of emergency risks. Let x = 0.5, z = 0.5, and β = 0.3, 0.6, 0.9 be the initial values (Figure 1a).*

**Scenario 2.** *The government selects the "reward-punishment" strategy, environmental NGOs take a neutral attitude, and logistics enterprises face different degrees of emergency risks. Let x = 1, z = 0.5 and β = 0.3, 0.6, 0.9 be the initial values (Figure 1b).*

**Scenario 3.** *The government selects the "reward-punishment" strategy, environmental NGOs choose the "participation" strategy, and logistics enterprises face different degrees of emergency risks. Let x = 1, z = 1 and β = 0.3, 0.6, 0.9 be the initial values (Figure 1c).*

**Scenario 4.** *The government takes a neutral attitude, environmental NGOs choosing the "participation" strategy, and logistics enterprises face different degrees of emergency risks. Let x = 0.5, z = 1 and β = 0.3, 0.6, 0.9 be the initial values (Figure 1d).*

Figure 1a shows that when both of the government and environmental NGOs held neutral attitudes towards emergency recycling processes, logistics enterprises' strategy choice will evolve towards the direction of "non-participation" regardless of the degree of emergency risks. However, if environmental NGOs devote themselves to supervising the government and logistics enterprises, and the risk of emergencies is low, logistics enterprises will eventually choose the "participation" strategy even if the government takes a neutral attitude (see Figure 1d), which indicates the participation of environmental NGOs can help the government optimize and improve the effect of supervision.

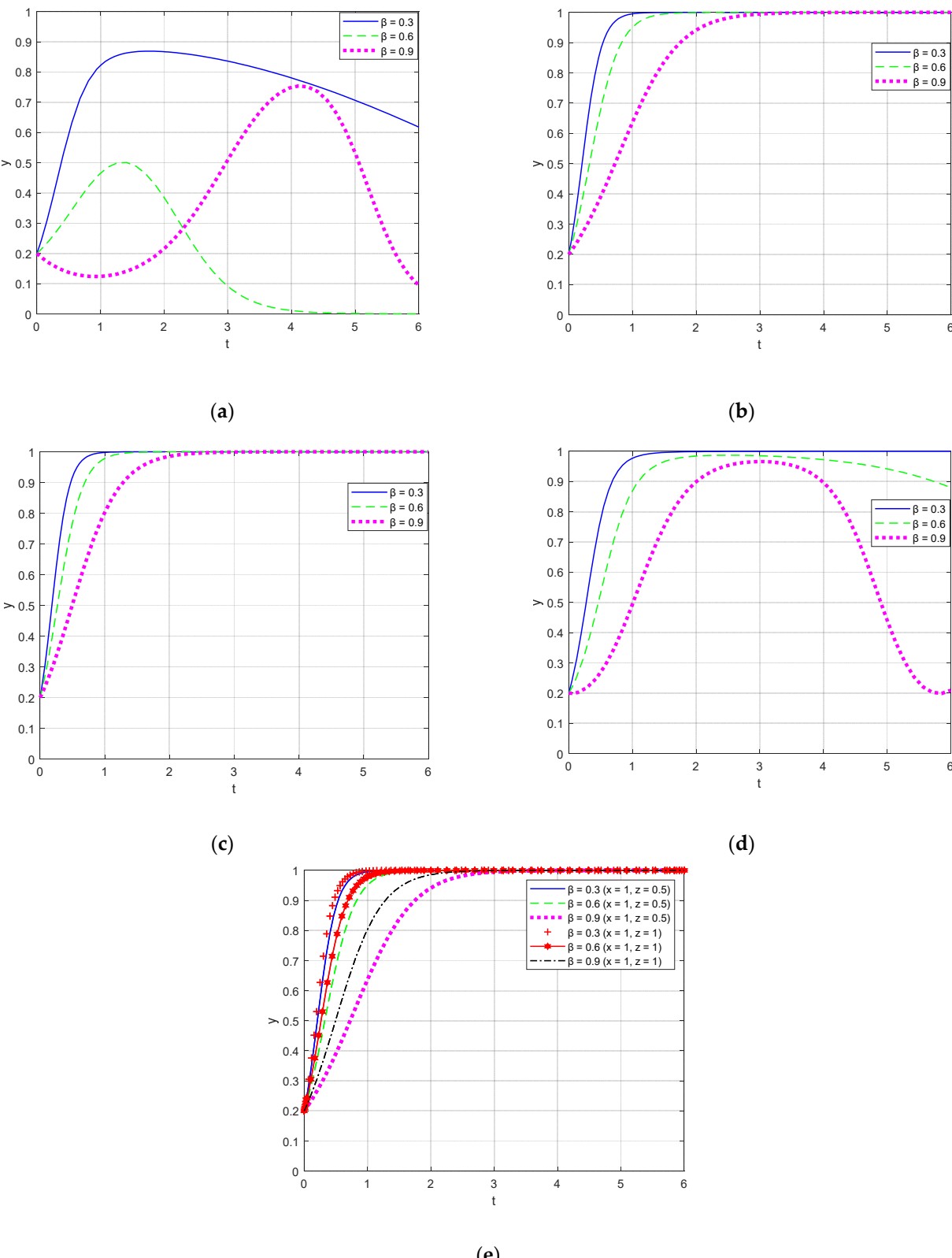

**Figure 1.** (**a**) The strategy evolution of logistics enterprises in Scenario 1; (**b**) The strategy evolution of logistics enterprises in Scenario 2; (**c**) The strategy evolution of logistics enterprises in Scenario 3; (**d**) The strategy evolution of logistics enterprises in Scenario 4; (**e**) Comparison of logistics enterprises' strategy evolution in Scenario 2 and Scenario 3.

Figure 1b,c illustrate that, regardless of the degrees of emergency risks, if the government chooses the "reward-punishment" strategy, the strategic choice of logistics enterprises will evolve towards the direction of "participation". In addition, as shown in Figure 1e, the logistics enterprise in Scenario 3 could reach a stable state faster than in Scenario 2. In other words, when the government is fully engaged in the regulation of emergency recycling processes, as well as environmental NGOs fully carrying out the function of supervision, logistics enterprises could reach evolutionary stability faster. The simulation results further prove that the government's choice of "reward-punishment" strategy is a sufficient condition for logistics enterprises to choose the "participation" strategy, that is, the relevant government intervention can directly promote logistics enterprises to participate in emergency recycling processes under the emergency risks.

*5.2. Impact of Environmental NGOs' Intervention on Logistics Enterprises in the Context of "Government Failure"*

"Government Failure" refers to various negative consequences in the governance of social issues due to the limitations of government's intervention, work mistakes or defects. It will eventually lead to inefficient government work and harm public welfare [13]. As for the evolutionary game model of this study, the government's "reward-punishment" strategy is able to promote logistics enterprises to implement reverse emergency logistics. However, there may also be "Government Failure" during the course of the government adopting the "reward-punishment" strategy. Thus, considering different degrees of "Government Failure", this study further simulates the impact of environmental NGOs' intervention on the strategic choice of logistics enterprises.

With initial values determined, let $x = 1$, indicating that the government has fully fulfilled the "reward-punishment" strategy of emergency recycling processes. Let ($C_{G2}$, $R_{G3}$) = ($0.0C_{G2}$, $0.0R_{G3}$), ($0.5C_{G2}$, $0.5R_{G3}$), and ($0.8C_{G2}$, $0.8R_{G3}$) = (0, 0), (2, 1), and (3.2, 1.6) to be three levels of "Government Failure", that is, 'complete', 'moderate', and 'low', respectively (see Figure 2).

As shown in Figure 2a, environmental NGOs must try their best to perform the supervision function so that logistics enterprises will choose the "participation" strategy, when the government chooses the "reward-punishment" strategy but there is a complete "Government Failure". Otherwise, the strategy choice of logistics enterprises will eventually evolve to the direction of "non-participation". In case of moderate and a low "Government Failure" (see Figure 2b,c, respectively), logistics enterprises will eventually choose the "participation" strategy as long as environmental NGOs fulfill their supervisory duties. Additionally, with the increasing probability of environmental NGOs selecting the "participation" strategy, logistics enterprises could reach the steady state faster. These results show that under the different levels of "Government Failure", the intervention of environmental NGOs is certainly able to improve the deficiencies caused by the "Failure" of government and further promote logistics enterprises to implement reverse emergency logistics.

*5.3. The Mutual Impact of the Governmental and Environmental NGOs' Strategy Choice*

5.3.1. The Impact of Environmental NGOs' Supervision on the Government's Strategy Choice

According to the evolutionary model designed above, besides alleviating "Government Failure", environmental NGOs involving in the emergency recycling processes also have an impact on the government's strategy. In order to test this hypothesis, this study further simulates these scenarios, and let $z = 1$ (indicating environmental NGOs choose "participation"), $x = 0.2$ (indicating the possibility of the government initially choose "reward-punishment" strategy is 0.2), $\Delta C = 4, 8, 12, 16, 20$ (indicating when NGOs choose "participation", the potential loss caused by their supervision to the inactive government in different degrees) (see Figure 3).

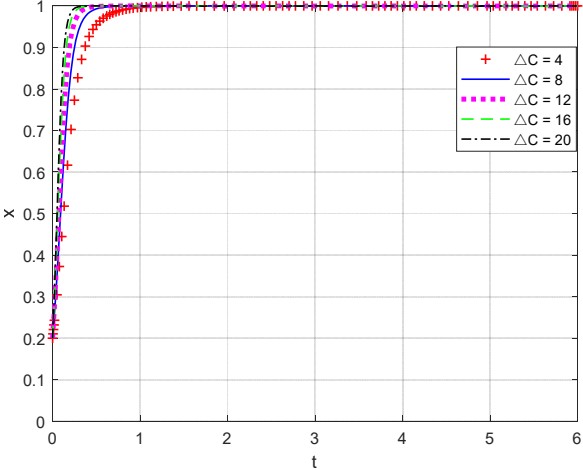

**Figure 2.** (**a**) The strategy evolution of logistics enterprises with complete degree of "Government Failure"; (**b**) The strategy evolution of logistics enterprises with moderate degree of "Government Failure"; (**c**) The strategy evolution of logistics enterprises with low degree of "Government Failure".

**Figure 3.** The impact of environmental NGOs' supervision on government's strategy choices.

As shown in Figure 3, although the possibility of the government initially choosing the "reward-punishment" strategy is low (e.g., 0.2), if environmental NGOs are completely involved in the emergency recycling processes, the government will eventually choose the "reward-punishment" strategy. In addition, the stronger the supervision of environmental NGOs, the faster the government will reach the stable state. The result also indicates that the relevant measures of environmental NGOs, such as generating public pressure and disclosing mistakes or defects to the upper authorities, can effectively promote the government to fully fulfil the regulation of emergency recycling processes.

### 5.3.2. The Impact of the Government's Support on Environmental NGOs' Strategy Choice

As mentioned above, environmental NGOs face uncertainties when they are performing their supervisory duties in the emergency recycling process. This article quantifies the impact of emergency uncertainties on environmental NGOs' strategy selection as an uncertainty cost coefficient $\gamma$ ($0 < \gamma < 1$). The three levels of the emergency uncertainty faced by environmental NGOs—low, medium, and high levels—could be set as $\gamma = 0.3$, $\gamma = 0.6$, and $\gamma = 0.9$, respectively. Then, the impacts of the government's support on environmental NGOs' decisions under different levels of uncertainty are simulated with initial parameter values (i.e., $z = 0.2$, $x = 1$, $\gamma = 0.3$, 0.6, and 0.9, respectively). As shown in Figure 4a, if the government fully implements the "reward-punishment" strategy (i.e., $x = 1$), the environmental NGOs will eventually choose the "participation" strategy regardless of the levels of uncertainty.

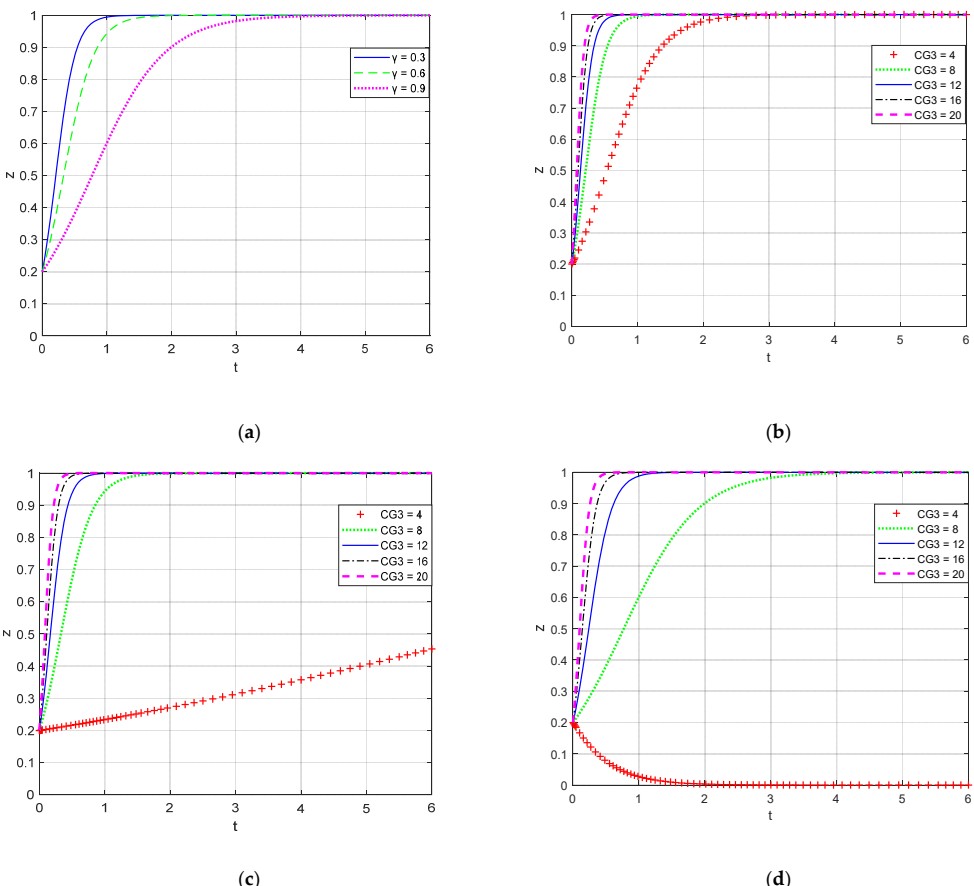

**Figure 4.** (**a**) The strategy evolution of environmental NGOs when the government chooses "reward-punishment"; (**b**) The impact of government incentives on the strategy evolution of environmental NGOs under low uncertainty; (**c**) The impact of government incentives on the strategy evolution of environmental NGOs under medium uncertainty; (**d**) The impact of government incentives on the strategy evolution of environmental NGOs under high uncertainty.

In addition, considering different levels of government support and uncertainties, how will the NGOs choose? Let $C_{G3}$ = 4, 8, 12, 16, and 20 to indicate the different levels of government support for environmental NGOs; let $\gamma$ = 0.3, $\gamma$ = 0.6, and $\gamma$ = 0.9 represent low, medium, and high levels of uncertainty. The results of the simulations are shown in Figure 4b–d, respectively. Figure 4c,d shows that environmental NGOs eventually evolve in the direction of "non-participation" under the medium and high levels of uncertainty if government gives relatively low incentives.

On the contrary, if the incentives given by the government are greater than a certain value, environmental NGOs will eventually choose the "participation" strategy regardless of the degree of uncertainty. Furthermore, as incentives or rewards increase, the speed at which environmental NGOs reach a stable state will be accelerated (see Figure 4b–d).

Together, these results imply that the support of the government could help environmental NGOs overcome uncertainties. With the support of the government, environmental NGOs could effectively and efficiently perform their supervisory duties, alleviate the phenomenon of "Government Failure", and urge the government to do its utmost to supervise the emergency recycling process. In addition, they can also assist the government to better promote logistics enterprises to participate in reverse emergency logistics.

## 6. Conclusions and Recommendations

### 6.1. Conclusions

In terms of the model of government-led and multi-participation emergency reverse logistics, this study constructs a tripartite evolutionary game model of the government, logistics enterprises, and environmental NGOs. Based on the above stability analysis of the three parties' strategy evolution, the results of the game showed that: (1) Choosing the "reward-punishment" strategy of government is a sufficient condition for the whole game system to reach a stable state. In other words, with any conditions, when the government chooses the "reward-punishment" strategy, the whole game system will reach a stable state; (2) the evolution of government strategy is related to its net losses; (3) under the condition that the government chooses the "reward-punishment" strategy, as long as environmental NGOs and logistics enterprises actively participate in emergency recycling processes, they can acquire profits; (4) the strategic evolution of logistics enterprises and environmental NGOs are related to their net benefits, and the net benefits are significantly related to the emergency risk cost coefficient ($\beta$) and uncertainty coefficient ($\gamma$).

Based on the ESS analysis results, we further investigated how the government and environmental NGOs influence the strategic evolution of logistics enterprises through simulation studies by using MATLAB R2018b software. In the simulation analysis, the emergency risk cost coefficient ($\beta$), uncertainty coefficient ($\gamma$), different levels of "Government Failure", and the intervention of environmental NGOs were considered. The results showed that (1) the government's full implementation of the "reward-punishment" strategy is a sufficient condition for logistics enterprises to implement reverse emergency logistics and for environmental NGOs to perform their duty of supervision; (2) the risk of emergency makes logistics enterprises take a passive attitude, while the government's "reward-punishment" strategy could directly promote logistics enterprises to carry out reverse emergency logistics, and the active supervision of environmental NGOs can also motivate logistics enterprises to participate in emergency recycling processes; (3) the intervention of environmental NGOs can effectively make up for the mistakes or defects caused by "Government Failure", thereby helping the government promote the logistics enterprises to implement reverse emergency logistics; (4) in addition to alleviating "Government Failure", the supervision of environmental NGOs can also stimulate the government to fully fulfill the "reward-punishment" strategy of emergency processes; (5) the uncertainties of an emergency will affect the participation enthusiasm of environmental NGOs, while government incentives and support can effectively promote environmental NGOs to perform their supervisory duties.

Nevertheless, although the research of this study is of certain practical significance, some limitations and deficiencies remain. (1) First, this paper only considers three participants (government, logistics enterprises and environmental NGOs) in emergency recycling processes. There may be more players involved in the game model in the context of a real emergency situation, including other NGOs, medical institutions, people in disaster areas, etc. Thus, subsequent research could involve more partners or try to analyze various game combinations of players. (2) Second, the research introduces the influence of emergency risks and uncertainties to the game system, but other impact factors were not considered. Further studies could investigate and identify more factors affecting the game model with the background of emergency recycling. (3) Third, this paper investigates different case scenarios according to the evolutionary game model by utilizing simulation data. In subsequent research, more real emergency recycling cases can be studied in depth.

*6.2. Recommendation*

Based on our findings, this paper proposes the following policy suggestions to the policy makers:

(1) The emergency risk can significantly influence the strategy choice of logistics enterprises. The emergency risk leads to the additional costs of logistics enterprises, and with the increase of it, the probability of enterprises implementing emergency reverse logistics decreases gradually. In this situation, logistics enterprises are unwilling to participate in the emergency recycling processes without sufficient external help. Therefore, in order to promote logistics enterprises' participation, the government must play the full role of regulation and guidance in different ways. On the one hand, the government could set rewards and compensate enterprises' losses incurred by emergency risk during the recycling processes. On the other hand, the government could also set punishments to increase the pressure on those passive logistics renterprises.

(2) It is necessary for environmental NGOs to be involved in the emergency recycling mechanism. In this study, we found that the intervention of environmental NGOs is able to support government to improve logistics enterprises' participation more efficiently and effectively. More important, though the government does mainly promote logistics enterprises to fulfill the main recycling affairs, "Government Failure" which reduces the effect of the regulation and guidance for logistics enterprises cannot be ignored. Environmental NGOs' intervention is capable of complementing the government's defects to promote logistics enterprises' participation under the various degrees of "Government Failure". Furthermore, the potential losses caused by NGOs' supervision measures plays a crucial role in the government's strategy selection.

(3) The uncertainties brought about by emergency situations and other game players' unpredictable behaviors can also impact the strategy choice of environmental NGOs. The uncertainty brings about additional costs to NGOs. Additionally, it is almost impossible for NGOs to involve into the emergency recycling mechanism under high uncertainties. According to the research results, with the financial support of the government, environmental NGOs are competent in performing supervisory duties better. Thus, the government ought to consider the long-term social and economic benefits of environmental NGOs and consider providing them with subsidies to overcome the negative influence of uncertainties during recycling processes.

All in all, policy makers should notice that (1) logistics enterprises are able to assume the responsibility of emergency reverse logistics; (2) environmental NGOs can not only help the government incentivize logistics enterprises' participation, but also could alleviate "Government Failure" and urge the government to do its utmost to supervise the emergency recycling process; (3) the fully active regulation and guidance of the government is the most essential factor in the entire emergency recycling mechanism.

**Author Contributions:** Conceptualization, Y.Z. and Y.L.; methodology, Y.Z. and Y.L.; software, Y.Z.; validation, Y.Z., Y.L. and L.Y.; writing—review and editing, Y.Z. and Y.L.; visualization, Y.Z. and L.Y.; supervision, Y.L. All authors have read and agreed to the published version of the manuscript.

**Funding:** This research was funded by the National Natural Science Foundation of China [71402159], Yunnan University Eastland Young and Middle-aged Backbone Teachers Training Program (2018), 2021 Yunnan Provincial Academic Degrees Committee Program [SJZYXWALK202113], Yunnan University Program [XJKCSZ202117] and [2022XJYZKC11].

**Institutional Review Board Statement:** Not applicable.

**Informed Consent Statement:** Not applicable.

**Data Availability Statement:** The simulation data and code that support the findings of this study are openly available in [figshare] at https://doi.org/10.6084/m9.figshare.20713576, accessed on 29 August 2022.

**Conflicts of Interest:** The authors declare no conflict of interest.

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
