# Peer review of "How to Promote Logistics Enterprises to Participate in Reverse Emergency Logistics: A Tripartite Evolutionary Game Analysis"

_sustainability, doi:10.3390/su141912132_

Round 1

Reviewer 1 Report

The paper is well-written and organized. The paper provides well-defined problem formulation and analysis. 

However, it will be difficult for decision-makers to adopt it. Thus, provide a case study so that decision-makers can adopt it easily. Or solve it using any optimizer (GA, PSO). You may refer following works for optimization problems.

An optimization model for reverse logistics network under stochastic environment by using genetic algorithm.

A genetic algorithm-based heuristic for the dynamic integrated forward/reverse logistics network for 3PLs.

Routing and scheduling optimization for UAV assisted delivery system: A hybrid approach

Reviewer 2 Report

Dear Authors

Thanks for submitting this manuscript aiming to investigates how to control the dynamic evolutionary relationship between the Government, Logistics Enterprises, and Environmental NGOs so that the emergency reverse logistics development evolves in a direction favorable to the interests of all three parties. To make it more valuable for the readers, I have some advice below:

Topic: An Evolutionary Game Analysis linking the Government, Logistics Enterprises, and Environmental NGOs in Reverse Emergency Logistics

Please clarify the novelty of this paper with respect to the published paper:

https://www.tandfonline.com/doi/abs/10.1080/01605682.2021.1880294?journalCode=tjor20

https://doi.org/10.1080/01605682.2021.1880294

Please explain if the article is new, and why is the title of the article so similar.

What is the difficulty of this article? Please summarize and state

Abstract

The abstract is not well written. in the abstract, you must pay briefly to the subject, practical problem / theoretical gap/contribution, object, methodology, and results/implications.

Introduction

The authors didn't give a good introduction. The Introduction section is too long. It is better to consider a new section that could be titled background/related works, there are a few typos that need to be corrected. Carefully read through this section to effect the corrections.

Row (86) to (98) are very similar to the mentioned article. Please rewrite them to avoid plagiarism.

Literature Review   

The authors have done very well on the theoretical underpinning of the study. However, the literature review section needs to be strengthened. The literature section is an integral part of the paper. In this paper, the review of the literature section is weak.  

 Methodology(The model)

Authors should check the article for typos and grammatical errors. In general, the typeset equations should be regarded as parts of a sentence and treated accordingly with the appropriate grammatical convention and punctuation. More editing for writing is needed. At the end of all equations must be put "COMMA" or "POINT" according to the typing rules. The author must use the latex journal form and use the correct latex relations.

Row (306) F(x)= ? It is unclear, please correct it. The same problem exists in a row (309).

Mathematical relations in a row (315), (316), and (317) are wrongly typed, Please correct it.

Row (383) seems incomprehensible, Please rewrite it again.

Be sure to standardize English expressions, and adjust page layout, paragraph indentation, and reference format.

Data Simulation 

This section is well written. However, there are a few typos that need to be corrected. Carefully read through this section to effect the corrections.

Conclusion 

Research/Practical limitations for following researchers need to be addressed in a separate paragraph at the end of the conclusion section, also check for grammar and sentence structure.

Overall, the study is a good piece of work but it should be endorsed for publication if these few comments are addressed.

Reviewer 3 Report

The manuscript investigates activity and how to control the dynamic evolutionary relationship between government regulators' supervision, logistics enterprises' operating behaviour and environmental NGOs. In general, this paper is well written and the topic is interesting. Here, there are some concerns of this reviewer:

Comments to the Authors:

1. Abstract:

I suggest the authors to define some of the terminologies such as NGOs first before abbreviation. 

2. Index Terms/ Keywords:

The keywords should be arranged in alphabetic order.

3. Literature Review

a)      Citation/Refs. are not properly presented. For example, Chen, Zhang [10], Fiedrich, Gehbauer [21], Jeong, Hong [27], Lu, Wu [28], Yu, Gao [29], Zhang, Su [30], etc., should be written as Chen et al. [10], Fiedrich et al. [21], ….

b)     Balcik and Beamon. [26], Pettit and Beresford. [32], etc., are not clear. I encourage the authors to remove the full stop before [26] and [32].

c)      In the 1950s, Herbert A Simon, the pioneer of behavioral decision theory, published a book entitled ‘Administrative Behavior’. This statement requires citation.

The model

a)      Page 7, lines 287 to 290: I encourage the authors to name/label the equations as (1), (2), etc.

b)     Same thing for lines 293 to 296, 299 to 302, line 306, line 309, etc.,
